# Self-Supervised Learning of Multi-Object Keypoints for Robotic Manipulation

Jan Ole von Hartz*, Eugenio Chisari*, Tim Welschehold, and Abhinav Valada

*Abstract*— In recent years, policy learning methods using either reinforcement or imitation have made significant progress. However, both techniques still suffer from being computationally expensive and requiring large amounts of training data. This problem is especially prevalent in real-world robotic manipulation tasks, where access to ground truth scene features is not available and policies are instead learned from raw camera observations. In this paper, we demonstrate the efficacy of learning image keypoints via the Dense Correspondence pretext task for downstream policy learning. Extending prior work to challenging multi-object scenes, we show that our model can be trained to deal with important problems in representation learning, primarily scale-invariance and occlusion. We evaluate our approach on diverse robot manipulation tasks, compare it to other visual representation learning approaches, and demonstrate its flexibility and effectiveness for sample-efficient policy learning.

## I. INTRODUCTION

Despite major advancements in reinforcement and imitation learning, sample efficiency is still a dominant challenge for both techniques, severely limiting their applicability to robotic manipulation. While learning manipulation policies from raw camera observations is typically computationally expensive and requires large amounts of training data, ground truth scene features are usually not available outside of simulation. This results in a large disparity between the potential promised by the state of art in e.g. reinforcement learning research and their practical use in robotic manipulation. Representation learning has been exploited to bridge the gap between training on camera observations versus ground truth features [1] and over the years, a wide variety of approaches have been proposed. However, not all of them are equally suited for the challenges of policy learning in robotic manipulation. From experience, we offer the following criteria: 1) Meaningfulness, i.e. to encode rich semantic content relevant to the task. 2) Compactness, to enable efficient policy learning from small amounts of data. 3) Invariance to image rotation, scale and partial occlusions, i.e. be temporally and spatially consistent. 4) Interpretability, to foster trust and safety. 5) Applicability to deformable objects and multi-object scenes. 6) Require minimal supervision.

One family of representation learning approaches are pose estimation-based methods [2]–[5]. Although being compact, meaningful and easy to interpret, they typically need a 3D model of the object, they are not applicable to deformable objects, and do not work well in the presence of occlusions. Image reconstruction-based methods [6]–[8], on the other

*These authors contributed equally.
Department of Computer Science, University of Freiburg, Germany.

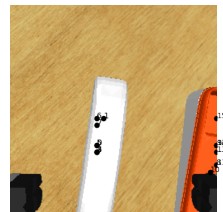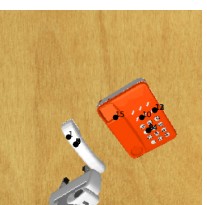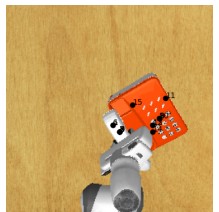

Fig. 1: We demonstrate that keypoints can be trained to be scale-invariant and handle occlusions, while tracking multiple objects across time and camera perspectives.

hand, are applicable to arbitrary scenes, but are hard to interpret and their usefulness for policy learning is limited due to the large discrepancy between pretext and downstream tasks. Interestingly, Dense Object Nets (DON) trained on dense correspondence were found to produce object keypoints suitable for robot manipulation [9]–[11]. Constituting a compact and easy to interpret representation, they facilitate efficient policy learning. Notably, they can even generalize across objects. Nevertheless, they have only been studied in settings without occlusions and moving cameras [9]–[12]. Moreover, in the context of policy learning, they have only been demonstrated to work for simple single-object tasks.

A major problem in image representation learning is scale-invariance, i.e. the ability to find corresponding image features across views with different distances between the object and the camera. Whereas, in the domain of handcrafted features, substantial work has been done to achieve scale-invariance [13], both work on keypoints [9]–[12] and other prior work [1] has avoided this problem by using fixed overhead cameras. However, this severely limits the utility of this method for diverse applications. For example, it can neither be used for mobile robots, nor for manipulation tasks where a wrist camera is required for precise alignment of the gripper.

In this work, we evaluate the feasibility of using keypoints as a representation for policy learning on a new set of simulated tasks including, for the first time, multi-object tasks. The selected tasks introduce a set of new challenges, most importantly, transparency, occlusion, moving cameras (hence the need for scale-invariance) and stereo correspondence. We perform extensive evaluations and demonstrate that DON-based keypoints can be trained to deal with all these challenges, often outperforming other methods.

In summary, our main contributions are:

1) We extend Dense Object Nets to deal with scale variances and occlusions, e.g. due to moving cameras.
2) We demonstrate that keypoints are useful for learning multi-object manipulation tasks, where additional objects are not just clutter, but relevant to the task.

3) Finally, comparing it to other representation learning approaches, we identify key challenges and propose potential solutions for future work.

## II. RELATED WORK

*Representation Learning*: A common approach to generate more compact representations of camera observations is by training a neural network model with a bottleneck such as a Variational Auto-Encoder (VAE) for image reconstruction. An example of this approach is $\beta$-VAE [7] which can be parameterized to favor disentangled representations. MONet [6] partitions the image into several *slots* first, that are subsequently encoded using multiple VAEs. In both cases, the latent representation at the respective bottleneck(s) serves as the representation for the downstream task. Similarly, Transporter [8] is trained to reconstruct a source image $I$ from a target image $I'$ via transporting local features. In [1], these representations have been compared and shown to enable more efficient policy learning on a set of robotic manipulation tasks. Due to their agnosticism toward the scene's content, in principle, all these methods work out of the box with multi-object scenes, occlusions and changes in perspective.

*Keypoints*: Keypoints are a set of pixel- or 3D coordinates, usually placed on the task-relevant objects, and that are ideally invariant to changes in the camera and object positions. These keypoints can be generated by training an encoder on the dense correspondence pretext task[9]. They are suited for policy learning via behavioral cloning [10] and model-based reinforcement learning (RL) [11]. Subsequent work has shown that keypoints can be learned end-to-end through RL [14]. Moreover, keypoints are able to generalize between instances of the same object class [9] and they are also applicable to deformable objects [10].

*Multi-Object Scenes*: Recent work [12] explores the use of DC-generated keypoints in a multi-object setting. The authors employ a similarity graph between scenes, from which they sample using random walker sampling. To label the similarity between scenes, they leverage a pretrained ResNet [15] and cluster the resulting embeddings. Strikingly, this allows them to retain within-class generalization, while simultaneously distinguishing object classes without additional supervision. However, this method has a number of drawbacks. First, it requires single-object scans of all objects. Second, as the authors noted, it deals poorly with occlusions. Third, their similarity measure requires the computation of a $K$-means clustering. This both leads to increasing complexity of the method, scaling with the number of training sequences and leads to further problems when the number of classes is not known. Much of this stems from attempting to generalize between objects of the same class. Florence *et al.* [9] identified that their underlying technique can be used for multi-object scenes, but did not provide a way to train the network to distinguish between the individual objects in a multi-object scene. In contrast, our method instead allows to collect data from multi-object scenes and to directly train the DON on them. This not only makes data collection much faster and removes the mentioned computational overhead, but also allows to directly train the encoder for object-discrimination and occlusions.

*Policy Learning*: Another common problem for visual representations is object scaling, i.e. finding correspondences between views of the same object that show it from different distances. Prior work circumvents this problem by either planning a grasp from a fixed height [9], [12] or, for policy learning, capturing the scene using a fixed overhead camera [10], [11]. This restricts the use of DONs, for example, with a moving wrist-camera. In robotic manipulation, however, such a camera might be needed for a precise grasping. Similarly, prior work does not address the problem of occlusion. In this work, we demonstrate that DONs can be trained to be invariant to scale as well as to occlusions, while still having the ability to distinguish multiple objects.

## III. TECHNICAL APPROACH

Our goal is to extract keypoints from camera observations for efficient policy learning. We achieve this by training a DON in a self-supervised manner, for which we first need to estimate ground truth pixel correspondences between pairs of images. In this section, we first recap the general DON pretraining procedure and keypoint generation. We then describe how to adapt these techniques to the multi-object case and how to achieve scale-invariance. Finally, we detail how we use the keypoints for policy learning.

### A. Correspondence Estimation

By moving a RGB-D camera in a static scene, we can reconstruct the 3D representation of that scene, e.g. using volumetric reconstruction [16], from which a point cloud can be extracted. After filtering out background points, we can project the point cloud back onto the image plane to generate object masks for all images along the trajectory. For a given pixel position in one image in the trajectory, we can find the corresponding pixel position in another image of the same trajectory via simple 3D projections using the respective camera pose and calibration matrix.

### B. Dense-Correspondence Pretraining

Using the aforementioned technique for finding correspondences between pairs of images, we train an encoder network $e_\eta : \mathbb{R}^{H \times W \times 3} \to \mathbb{R}^{H \times W \times D}$, mapping an RGB image to a $D$-dimensional descriptor, to minimize the descriptor distance between corresponding points, while enforcing at least a margin $M$ between non-corresponding points. In doing so, we utilize a self-supervised pixelwise contrastive loss [17], [18]. Specifically, for a given pair of images $I_a, I_b$, we sample $m$ pixel locations $u_a$ from the object mask of $I_a$ and compute the corresponding positions $u_b$ in $I_b$. Additionally, for each point $i$ in $u_a$ we sample $n$ non-corresponding points $u_i$ from both $I_b$'s object mask and the background. We then compute a gradient-update for the encoder as

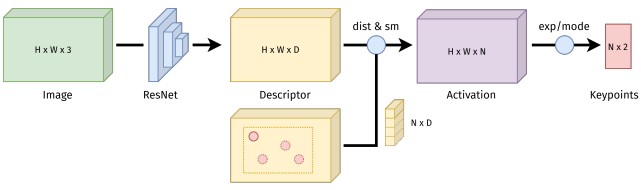

Fig. 2: Keypoint generation during policy learning.

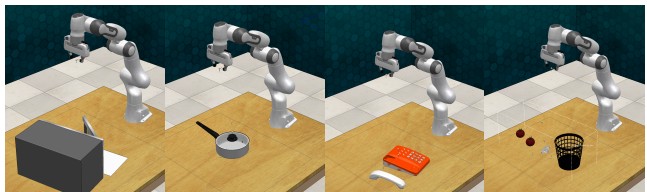

Fig. 3: RLBench Tasks: CloseMicrowave, TakeLidOffSaucePan, PhoneOn-Base, PutRubbishInBin

$$\mathcal{L}(I_a, I_b) = \sum_{i=0}^{m} \left( \frac{\|e_\eta(I_a)_{u_a^i} - e_\eta(I_b)_{u_b^i}\|^2}{m} \right.$$
$$\left. + \sum_{j=0}^{n} \frac{\max\left(0, M - \|e_\eta(I_a)_{u_a^i} - e_\eta(I_b)_{u_i^j}\|^2\right)}{n} \right). \quad (1)$$

Similar to [9], we use a ResNet50 encoder with stride 8 pretrained on ImageNet and bilinearly upsample the feature maps back to the full input resolution. We use the Adam [19] optimizer with an initial learning rate of $1e-4$ and exponentially decay by a factor of $0.9$ every 25 steps. Additionally, we regularize the training via a L2 penalty of $1e-4$.

### C. Keypoint Generation

During policy learning, we freeze the encoder and sample one frame from the set of trajectories that we want the policy to learn. We then either randomly sample reference positions from the relevant object masks or manually select them. The descriptors at these positions of the sampled image serve as the *reference descriptors* for the model. A camera observation is encoded by feeding it through the frozen encoder and computing the Euclidean distance between each of the reference descriptors and the respective descriptor at all positions in the embedding of the image. Applying a softmax to the negative distance map yields an activation map interpreted as the probability of each pixel location corresponding to the reference position. Unlike prior work [9], [10], we select the keypoint location as the global mode of the activation map, which we found to work better than the expectation in the presence of noise and multimodality. Fig. 2 illustrates this approach.

To generate 3D keypoints, [10] propose to project the pixel-coordinates into the world frame. In contrast, we find that either projecting the pixel-coordinates into the local camera frame, or even just appending the respective depth values to the coordinate vector, besides being simpler, yields a more effective representation for our LSTM policy. This is due the LSTM being sensitive to the scale of the data. We normalize the pixel coordinates to lie within $[-1, 1]$ to ease learning the policy.

### D. Scale-Invariance and Multi-Object Tasks

To extend DONs to multi-object tasks, we want to pretrain directly on multi-object scenes, such that the data is fast to collect and already contains occlusions. Thus, we need to generate separate masks for each object in the scene. To do so, we can employ volumetric reconstruction and split the resulting point cloud using simple clustering. Projecting these object-wise point clouds back onto the camera planes yields consistent object masks for the trajectory. During one iteration of pretraining, we sample one of the object masks and treat the other object as part of the background, teaching the model to distinguish the two objects. Just sampling an object mask has the added benefit of working with any number of objects and empty masks are skipped. Furthermore, we collect an additional set of trajectories, only showing the robot arm, to teach the model not to confuse it with the objects in the scene.

Similarly, as we found the DON to generalize badly to unseen perspectives such as close-ups, we needed to pretrain the model on similar perspectives it would see during policy learning. Note, that this is not limited to cases where the change in perspective cuts away necessary context, but to changes in distance between object and camera in general. In contrast, having these different perspectives in the training data teaches the model to generate a scale-invariant representation. In our experience, larger descriptor dimensions enable training the encoder on more perspectives without loss in quality. Yet, we find it important to normalize the descriptor distances in pretraining by the square root of the descriptor dimension. During policy learning, we sample an equal number of reference positions from all the object masks.

### E. Imitation Learning

We follow [20] in the setup of our experiments, with the action space being constituted by the change in the robot's end-effector pose. Using an LSTM, we predict the mean of a Gaussian action-distribution with fixed variance, i.e. $\pi_\theta \sim \mathcal{N}(f_\theta(s, \theta); \sigma^2)$. The variance is set to correspond to $1\,\text{mm}$ for the translational and $0.25$ degrees for the rotational components of the action. For the observation, we concatenate the visual representation with the robot's current joint angles or the end-effector pose. Across all trajectories contained in a batch and their time steps, we minimize the negative log-likelihood of the predicted action distribution as

$$\mathcal{L}(s, a) = -q \log(\pi_\theta(a \mid s)). \quad (2)$$

We again train the model using the Adam optimizer with a learning rate of $3e-4$ and an L2 penalty of $3e-6$.

## IV. EXPERIMENTAL EVALUATIONS

We evaluate the utility of different representations for policy learning using RLBench [21], a suite of realistic manipulation tasks using everyday objects. In this framework, between instances of the same tasks, the objects are placed randomly in the scene. We select two single-object tasks (CloseMicrowave, TakeLidOffSaucePan), two multi-object tasks (PhoneOnBase, PutRubbishInBin) and perform all of them with the model

TABLE I: Success rates of the learned policies.

| Method | Microwave | Lid | Phone | Rubbish |
|---|---|---|---|---|
| CNN | 0.615 | 0.315 | 0.420 | 0.245 |
| CNND | 0.560 | 0.180 | - | - |
| $\beta$-VAE [7] | 0.110 | 0.000 | 0.005 | 0.000 |
| Transporter [8] | 0.035 | 0.075 | 0.000 | 0.005 |
| MONet [6] | 0.785 | **0.875** | 0.385 | **0.760** |
| DC KP 2D | 0.805 | 0.280 | - | - |
| DC KP 3D | **0.935** | 0.800 | **0.640** | 0.590 |
| GT KP | 0.875 | **0.990** | **0.720** | **0.885** |

of a Franka Emika Panda robot, see Fig. 3. These tasks pose different challenges. In the CloseMicrowave task, we confront the models with an object that changes its shape and has very different appearance across a trajectory and in TakeLidOffSaucepan there is high object symmetry and transparency. The PhoneOnBase task requires careful alignment of the gripper and the PutRubbishInBin task adds visual clutter. The multi-object tasks further introduce occlusions and the need to track multiple objects. For the single-object tasks, we train the policy on 14 expert demonstrations, using a wrist-mounted camera with $256 \times 256$ pixels, while for the multi-object tasks providing 140 demonstrations and using a stereo setup of overhead and wrist camera (with identical resolutions). The overhead camera provides an overview over the scene, while the wrist camera facilitates alignment of the gripper with the objects and requires scale-invariance from the encoders. Both camera observations are encoded independently with the same encoder.

Besides the three pretrained representation learning methods $\beta$-VAE [7], Transporter [8] and MONet [6], we further compare against an end-to-end optimized Convolutional Neural Network (CNN) and a variant that has access to the camera's depth values (CNND). To disentangle the effects of representation and policy learning, we also add a ground truth keypoints model (GT-KP). We train all the policies for 1000 steps for the single-object tasks and 1500 steps for the multi-object tasks. We then evaluate all the policies in the respective task environments for 200 episodes.

### A. Single-Object Tasks

From the results shown in Tab. I, we observe that both $\beta$-VAE and Transporter learn representations that are unsuited for the tasks at hand. Transporter in particular is designed for scenarios with a top-down view on the scene and well-separated local features. Although the CNNs achieve reasonable policy success on CloseMicrowave, they are vastly outperformed by both Monet and Keypoints. Note that in TakeLidOffSaucePan, there is little information in the object's pixel-position but instead most of the information is in the camera depth. This can account for the performance gap between the 2D and 3D keypoints in Tab. I. Therefore, we drop the 2D keypoints and the underperforming CNND from the comparisons for the more difficult tasks.

While MONet manages to outperform the learned keypoints on TakeLidOffSaucePan, the GT-KP model still outperforms it by a large margin, indicating that keypoints are the more effective representation, although the current implementation

still has room for improvement. MONet's strong performance on TakeLidOffSaucePan is due to it partitioning the lid into its parts. It considers the dark parts (handle and knob) as one entity, thus enabling the policy to learn a sure grasp, and incidentally ignoring the difficulties entailed by the transparency of the rest of the lid.

### B. Multi-Object Tasks

In PhoneOnBase, many task instances are so difficult, that also the human pilot providing the demonstrations could only solve $84\%$ of the instances. Even among the remaining instances, a sizable fraction requires a more complex trajectory than the rest, such that the ground truth model's success rate of $72\%$ is close to what can be expected to be achieved by BC. While the learned keypoints come very close to this upper-bound on PhoneOnBase, the gap is larger on PutRubbishInBin due to the additional visual clutter and complex shape of the bin. MONet tends to conflate the robot arm with the scene object, making it perform poorly on PhoneOnBase, where precise alignment is critical. On PutRubbishInBin, where precision is less crucial, it outperforms the learned keypoints.

### V. CONCLUSIONS

Keypoints allow for efficient policy learning. Not only are they a compact representation that still encodes the task-relevant information, but if properly normalized, they are easy to learn from. Compared to other methods, this allows higher efficacy of the downstream policy when the training data is scarce, but precision is still required. Unlike CNNs and representations such as MONet, they can easily incorporate depth information, making them a powerful choice for robotic manipulation. Moreover, the method can be extended to new challenges in a straightforward manner. This includes, but is not limited to, multi-object tasks, occlusions, and changing object scale. While it needs to be specifically trained to handle these problems, doing so is intuitive, making the method a flexible approach. Compared to, e.g. MONet, keypoints generalize less well to unseen camera perspectives, and *need* to be specifically trained for new challenges, whereas MONet is more robust and generalizes better out of the box. Even with the improved pretraining, keypoints remain significantly more noisy for the moving wrist camera than for the stable overhead camera.

The descriptor distance provides an intuitive measure of the encoder's certainty that can be used in multiple ways, e.g. to ignore or resample uncertain keypoints to better deal with occlusions. Leveraging this fact will help to make the encoding less noisy. Moreover, using additional techniques such as random crop [22] in DC pretraining can help reduce the noise further. A more fundamental challenge lies in object symmetry when not enough context is provided to uniquely identify positions on the object. This is especially prevalent for the wrist camera, e.g. with the lid or phone receiver. One way to remedy this would be to introduce a memory component or extend the current camera observation by previously seen frames to add context. Finally, pretraining a Dense Object Net on a large set of different objects can enable the method to generalize beyond different instances of the same class.

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
