# OpenReview forum: "Self-Supervised Learning of Multi-Object Keypoints for Robotic Manipulation"
_ICRA.org/2022/Workshop/Contact-Rich — ICRA 2022 Workshop: RL for Manipulation Poster_

### Official Review · Reviewer_h4Tg · 2022-05-05
**Review of "Self-Supervised Learning of Multi-Object Keypoints for Robotic Manipulation"**

**Rating:** 8
**Confidence:** 4

**Review:**

Overall:
In this paper, the authors proposed to use keypoints as a representation for downstream policy learning. The major contribution of this work is to extend keypoint extraction to multi-object scenarios and demonstrate the effectiveness of the method in several challenging tasks.

1. This paper is clearly written and easy to understand. The effort of extending Dense Object Nets (DON) to multi-object setting is technically sound and well-motivated.
2. The authors perform extensive validation of the proposed approach in several challenging tasks compared to multiple baselines. The keypoint representation outperforms other representations in terms of policy performance.
3. The authors only use behavior cloning as the policy learning method. It would be good to see how the representations perform in other settings, e.g., RL and IRL.
4. The authors claim their proposed approach can work in object occlusion setting. However, DON only takes RGB images as input. I wonder how the method would work well when occlusion exists. It would be good if the authors can provide more explanation and insights about this.

---

### Official Review · Reviewer_YK8E · 2022-05-08
**Review of Paper "Self-Supervised Learning of Multi-Object Keypoints for Robotic Manipulation"**

**Rating:** 7
**Confidence:** 4

**Review:**

**Summary**: This paper explores the use of keypoints as the representation for policy learning. The keypoints are generated by the Dense Object Net (DON), which is pre-trained on dense correspondence tasks. The efficacy of the keypoints representation for policy learning is evaluated on four robot manipulation tasks, including two single-object tasks and two multi-objects tasks. The policies are trained by imitation learning. The method is compared with five baselines consisting of three pretrained representations and two end-to-end training baselines.

The main contribution of the paper is an extension to the training procedure of DON to address the problem of scale invariance, occlusion, and multi-objects scene.

**Strength**: The paper is clear and well-written. The considered robot manipulation tasks are challenging. The experimental results are quite good and provide insights on the advantages and disadvantages of different visual representations.

**Weakness**:  In Table 1, on the Microwave task, the ground truth keypoints model performs worse than the keypoint 3D model. This is strange because the ground truth keypoints model should provide the upper bound performance for the other keypoints-based models. Given more demonstrations, would the performance of the ground truth model eventually outperform, or at least as good as the keypoints-based model? Or are there any other reasons?